# DSCR1 deficiency ameliorates the Aβ pathology of Alzheimer's disease by enhancing microglial activity

Chiyeol Choi*, Hyerin Kim*, Jiyoung Oh, Chanho Park, Min Kim, Chu-Sook Kim, Jiyoung Park ⓘ

**Microglial phagocytosis and clearance are important for the removal of amyloid-β (Aβ) plaques in Alzheimer's disease (AD). Chronic exposure of microglia to Aβ plaques leads to microglial metabolic dysfunction, and dysregulation of microglia can accelerate the deposition of Aβ plaques and cause learning and memory impairment. Thus, regulating microglial Aβ clearance is crucial for the development of therapeutics for AD-related dementia. Here, Down syndrome critical region 1 (DSCR1) deficiency ameliorated Aβ plaque deposition in the 5xFAD mouse model of AD by altering microglial activity; however, the Aβ synthesis pathway was not affected. DSCR1 deficiency improved spatial learning and memory impairment in 5xFAD mice. Furthermore, DSCR1-deficient microglia exhibited accelerated lysosomal degradation of Aβ after phagocytosis, and BV2 cells with stable knockdown of DSCR1 demonstrated enhanced lysosomal activity. RNA-sequencing analysis showed that the transcriptional signatures associated with responses to IFN-γ were significantly upregulated in DSCR1-knockdown BV2 cells treated with Aβ. Our data strongly suggest that DSCR1 is a critical mediator of microglial degradation of amyloid plaques and a new potential microglial therapeutic target in AD.**

## Introduction

Alzheimer's disease (AD), the most common cause of dementia, is characterized by the accumulation of amyloid-β (Aβ) plaques and neurofibrillary tangles in the brain (Kumar et al, 2015; Pospich & Raunser, 2017). Aβ (1-42) is derived from the sequential cleavage of amyloid precursor protein (APP) by β- and γ-secretase, and failure of Aβ clearance results in aggregation of Aβ in the brain parenchyma (LaFerla et al, 2007; Karran et al, 2011). As the formation of Aβ plaques in the brain leads to neuronal dysfunction and death, proper clearance of Aβ is important for the treatment of AD (Karran et al, 2011).

Microglia are CNS-resident macrophages that engulf and degrade dead cells and various pathogens, including Aβ (Li & Barres, 2018; Damisah et al, 2020). Microglial activity has emerged as a key factor in neurodegenerative diseases (Hickman et al, 2018). Chronic exposure of microglia to Aβ plaques causes neuroinflammation and microglial dysfunction in AD (Baik et al, 2019; Leng & Edison, 2021). In contrast, restoration of impaired microglial activity recovers AD pathogenesis, including Aβ deposition, neuronal death, and cognitive defects (Baik et al, 2019). Therefore, targeting the modulation of microglial degradation activity could be an effective strategy to treat AD and AD-related dementia; however, the factors involved in regulating microglial clearance activity are not well understood.

Down syndrome critical region 1 (*DSCR1*), also known as regulator of calcineurin 1, is encoded on human chromosome 21 and mouse chromosome 16; it contains two functional domains: an RNA recognition motif domain and a calcineurin binding domain (Fuentes et al, 2000; Strippoli et al, 2000; Mehta et al, 2009). The role of DSCR1 as an inhibitor of CaN signaling has been well studied in various tissues (Fuentes et al, 2000; Casas et al, 2001; Wang et al, 2002), and CaN is also an important factor in T-cell activation (Aubareda et al, 2006). Previous studies have suggested a positive correlation between DSCR1 expression and AD pathogenesis, as DSCR1 expression is upregulated in the brains of patients with AD (Ermak et al, 2001; Harris et al, 2007). However, another study showed that upregulation of the nebular DSCR1 homologue in flies protects against APP-induced neurodegeneration (Shaw & Chang, 2013). Thus, how DSCR1 affects AD pathology is still controversial and remains to be investigated.

Here, we demonstrate that DSCR1 deficiency enhances microglial degradation of Aβ and reduces Aβ pathology, which improves impaired learning and memory in the 5xFAD mouse model. DSCR1 knockout (DSCR1 KO) microglia and DSCR1-knockdown (DSCR1-KD) BV2 cell lines were tested to examine lysosomal activity, and we found strikingly improved lysosomal activity in DSCR1-deficient microglia. Thus, these studies provide novel insights into the role of DSCR1 in microglia for AD therapy.

---

Department of Biological Sciences, College of Information and Bioengineering, Ulsan National Institute of Science and Technology, Ulsan, Republic of Korea

Correspondence: jpark@unist.ac.kr
*Chiyeol Choi and Hyerin Kim contributed equally to this work

# Results and Discussion

## DSCR1 deficiency ameliorates deposition of Aβ plaques, spatial learning, and memory impairment in 5xFAD mice

To investigate the function of *DSCR1* in AD, we crossed DSCR1 KO mice with 5xFAD mice (DSCR1 KO/5xFAD mice). First, we demonstrated the difference in Aβ plaque deposition between 5xFAD and DSCR1 KO/5xFAD mice through immunohistochemistry analyses of all brains. Surprisingly, DSCR1 deficiency significantly ameliorated Aβ plaque deposition in 5xFAD mice (Fig 1A). The occupied areas of the Aβ plaques decreased in the whole brains of the DSCR1 KO/5xFAD mice (Fig 1B) and in the hippocampus and cortex (Fig 1C and D) regions. In addition, DSCR1 KO/5xFAD mice exhibited a reduced number of amyloid plaques in these regions (Fig 1E–G), and the sizes of these plaques were reduced in the whole brain and cortex but not in the hippocampus of these mice (Fig 1H–J). Next, we showed significantly fewer dense core plaques in the DSCR1 KO/5xFAD mice by staining with Thioflavin-S (Fig 1K–N).

According to the Aβ hypothesis of AD (Karran et al, 2011), Aβ plaque deposition may lead to AD-related pathologies, including learning and memory impairments in mouse models (Selkoe & Hardy, 2016). Furthermore, recent studies have reported that reduced Aβ plaque deposition enhances AD-related cognitive deficits (Kim et al, 2015; Baik et al, 2019). Therefore, we hypothesized that DSCR1 deficiency could improve Aβ-induced spatial learning and memory impairment, and Aβ plaque deposition in 5×FAD mice. To verify this hypothesis, we performed the Morris water maze (MWM) test and found that DSCR1 KO/5xFAD mice showed restored spatial learning and memory in the escape latency (Fig 1O) and probe trials (Fig 1P and Q) of the MWM. Collectively, these data indicate that DSCR1 deficiency improves plaque pathology in the brain parenchyma and mitigates the impaired spatial learning and memory of 5xFAD mice.

## Amelioration of Aβ deposition in DSCR1 KO/5xFAD mice is not mediated by the Aβ synthesis pathway

There are two potential methods to reduce Aβ plaque deposition. The first is the reduction in Aβ synthesis (Suh et al, 2019), and the second is the improvement in Aβ clearance (Wang et al, 2017). With regard to Aβ synthesis, a decreased expression of APP or a decline in APP processing might be the cause of the amelioration in Aβ deposition. In addition, the sequential cleavage of APP by β- and γ-secretase influences the amyloidogenic pathway (Aβ) (LaFerla et al, 2007; Suh et al, 2019). Therefore, we examined human APP (hAPP) expression levels in 5xFAD and DSCR1 KO/5xFAD mice. We specifically analyzed hippocampal gene expression because the hippocampus is an important region for spatial learning and memory (Broadbent et al, 2004). Both mRNA and protein expression levels of APP were not significantly different in DSCR1 KO/5xFAD mice compared with 5xFAD mice (Fig 2A–D).

To further explore potential differences in the in vivo half-life of hAPP, we examined the half-life of hAPP in the brain of 5xFAD mice and DSCR1 KO/5xFAD mice for different durations. We showed that the turnover of hAPP was comparable among the two groups of mice (5xFAD mice versus DSCR1 KO/5xFAD mice) (Fig S1). Next, we examined

the expression level of BACE1 (β-secretase) to determine whether the proteolytic cleavage of APP by β-amyloid differs in DSCR1 KO/5xFAD mice (Suh et al, 2019). The results showed that the BACE1 expression level in DSCR1 KO/5xFAD mice was not significantly different from that in control and 5xFAD mice (Fig 2E and F). Together, these data revealed that the deletion of *DSCR1* had no effect on the Aβ synthesis pathway in 5xFAD mice. Therefore, the observed reduction in Aβ plaques in DSCR1 KO/5xFAD mice was not due to decreased APP production; rather, it was likely mediated by an enhanced Aβ clearance pathway.

## Microglia from DSCR1 KO mice degrade Aβ plaques faster than microglia from WT mice

How does lack of DSCR1 impact Aβ clearance? DSCR1 might influence several Aβ clearance mechanisms in the brain via proteolytic degradation or through the blood–brain barrier (BBB), interstitial fluid bulk flow, and cerebrospinal fluid egress pathways (Wang et al, 2017). First, we confirmed that DSCR1 ablation showed a limited effect on BBB permeability using a fluorescence-labeled dextran permeability assay (Fig S2). Considering that previous studies showed that clearance of Aβ was especially improved by microglial activation (Unger, 1998), we tested the effect of DSCR1 deficiency on microglia. We found that DSCR1 was mainly distributed in the whole cell area of primary microglia, and protein expression of DSCR1 was markedly down-regulated by fAβ treatment for 24 h (Fig S3). Next, we investigated plaque-associated microglial activity by immunostaining for MOAB2 and Iba1 (Fig 3A). We analyzed the plaque-associated microglia that were present around plaques of similar sizes in 5xFAD mice and DSCR1 KO/5xFAD mice to minimize the misinterpretation that larger plaque sizes lead to increased microglial accumulation (Jung et al, 2015). In addition, we investigated the number of plaque-associated microglia (Baik et al, 2019), microgliosis (Wang et al, 2015), and Iba1 intensity (Ito et al, 1998; Rodriguez et al, 2014) in the microglia. We found that the number of Iba1-positive microglia associated with plaque and microgliosis was not altered in the DSCR1 KO/5xFAD mice (Fig 3B and C). However, the Iba1 intensity was significantly higher in the DSCR1 KO/5xFAD mice than in the 5xFAD mice (Fig 3D), and we also noted up-regulated TREM2 expression around the plaques in the DSCR1 KO/5xFAD mice (Fig 3E and F), implying that the microglia in the DSCR1 KO/5xFAD mice were more activated than in the AD mice.

We cultured primary microglia from WT and DSCR1 KO mice and treated them with HiLyte Fluor 555–labeled fAβ. Subsequently, engulfment and degradation assays were performed to analyze microglial activity in response to Aβ. The engulfment assay indicated that fAβ engulfment was not significantly different among primary microglia from both mice at all incubation time points (5 min, 30 min, and 1 h) (Fig 3G and H). We then tested the Aβ clearance to measure microglial fAβ degradation (Paresce et al, 1997). Interestingly, the engulfed fAβ intensity declined more quickly in the primary microglia from DSCR1 KO mice than from WT mice at 1 and 2 d after washing the residual fAβ in media (Fig 3I and J), suggesting that fAβ was degraded faster in DSCR1-deficient microglia than in WT microglia. These results imply that the amelioration of Aβ plaque deposition in the DSCR1 KO mice/5xFAD mice was mediated by improved microglial Aβ degradation.

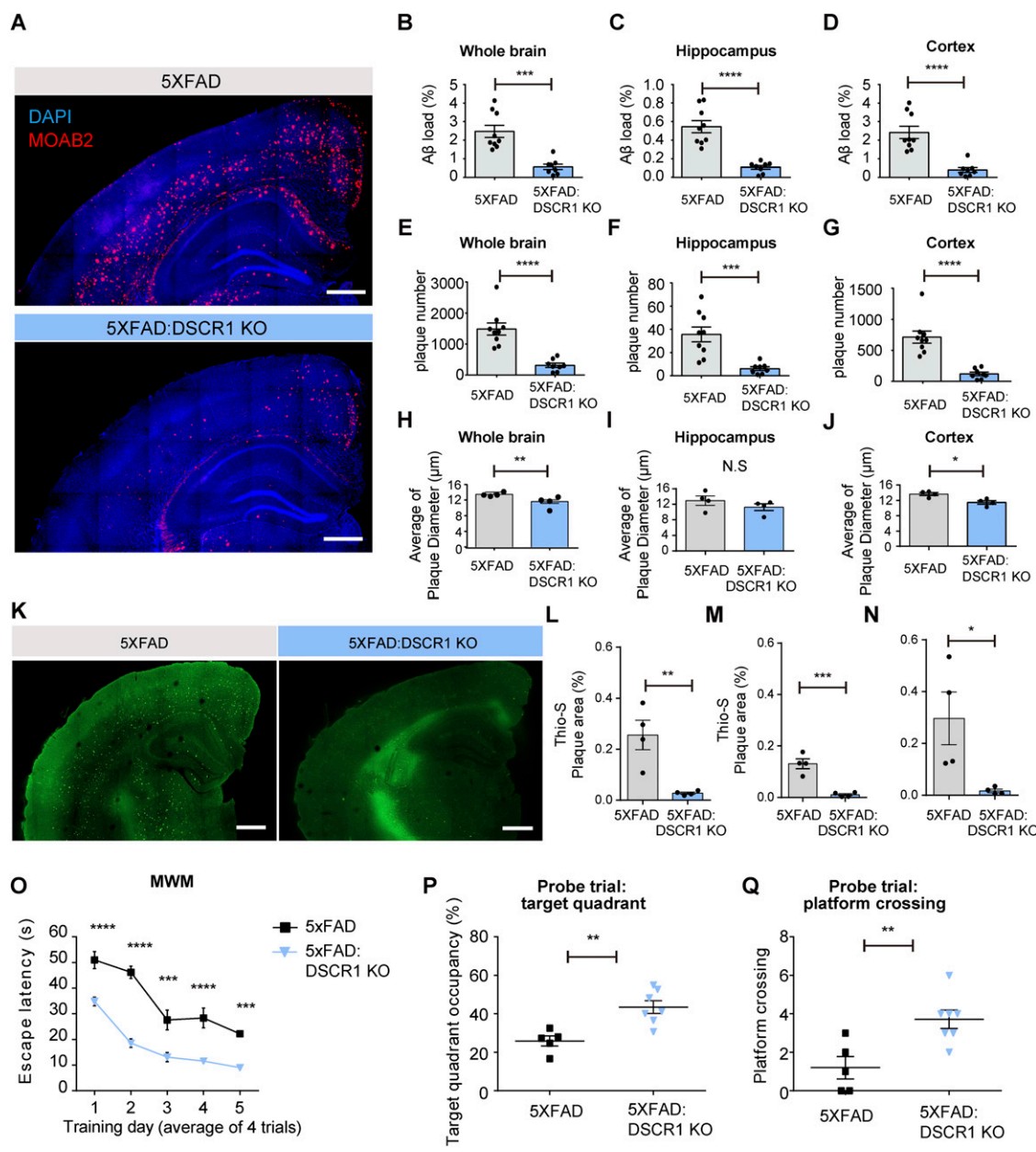

**Figure 1. DSCR1 KO ameliorated the deposition of Aβ plaques and impaired spatial learning and memory in 5xFAD mice.**
**(A)** Representative immunohistochemistry staining of Aβ (MOAB2) in 5-mo-old 5xFAD and DSCR1 KO/5xFAD mice. Scale bar = 500 μm. **(B, C, D)** Comparison of Aβ loads in the whole brain (B), hippocampus (C), and cortex (D) regions between 5xFAD and DSCR1 KO/5xFAD mice. **(E, F, G)** Total number of amyloid plaques between 5xFAD and DSCR1 KO/5xFAD mice in the whole brain (E), hippocampus (F), and cortex (G) regions (5xFAD, n = 9; DSCR1 KO/5xFAD mice, n = 8). **(H, I, J)** Quantification of amyloid plaque size in 5xFAD and DSCR1 KO/5xFAD mice in the whole brain (H), hippocampus (I), and cortex (J) regions (5xFAD; DSCR1 KO/5xFAD mice, n = 4). **(K)** Representative images of Thioflavin–S–positive Aβ plaques in 5xFAD and DSCR1 KO/5xFAD mice at 5 mo. Scale bar = 500 μm. **(L, M, N)** Quantitative analysis of Thio–S–positive amyloid plaque areas in 5xFAD and DSCR1 KO/5xFAD mice in the whole brain (L), hippocampus (M), and cortex (N) regions. (5xFAD; DSCR1 KO/5xFAD mice, n = 4) **(O)** Escape latency of the 5xFAD and DSCR1 KO/5xFAD mice was measured over five training days in the Morris water maze. **(P, Q)** Probe trials were performed on the last Morris water maze training day. **(P, Q)** Target quadrant occupancy (P) and platform crossing events (Q) were analyzed (5xFAD, n = 5; DSCR1 KO/5xFAD, n = 7). **(B, C, D, E, F, G, H, I, J, L, M, N)** Values are shown as the mean ± SEM and were tested for statistical significance using a two-tailed unpaired t test. N.S., not significant, *P < 0.05, **P < 0.01, ***P < 0.001, and ****P < 0.0001. **(O, P, Q)** Values are shown as the mean ± SEM. **(O)** Values were tested for statistical significance using repeated-measures two-way ANOVA. ***P < 0.001 and ****P < 0.0001. **(P, Q)** Values were tested for statistical significance using a two-tailed unpaired t test. **P < 0.01.

## Lysosomal activity is improved in DSCR1-deficient microglia

Lysosomes are major cellular organelles involved in the degradation of biomolecules and have emerged as therapeutic targets for various diseases (Bonam et al, 2019). Furthermore, microglial lysosomal activity has been reported to be an important factor for Aβ clearance (Majumdar et al, 2007; Majumdar et al, 2011). To prove that the improvement in microglial degradation was mediated by

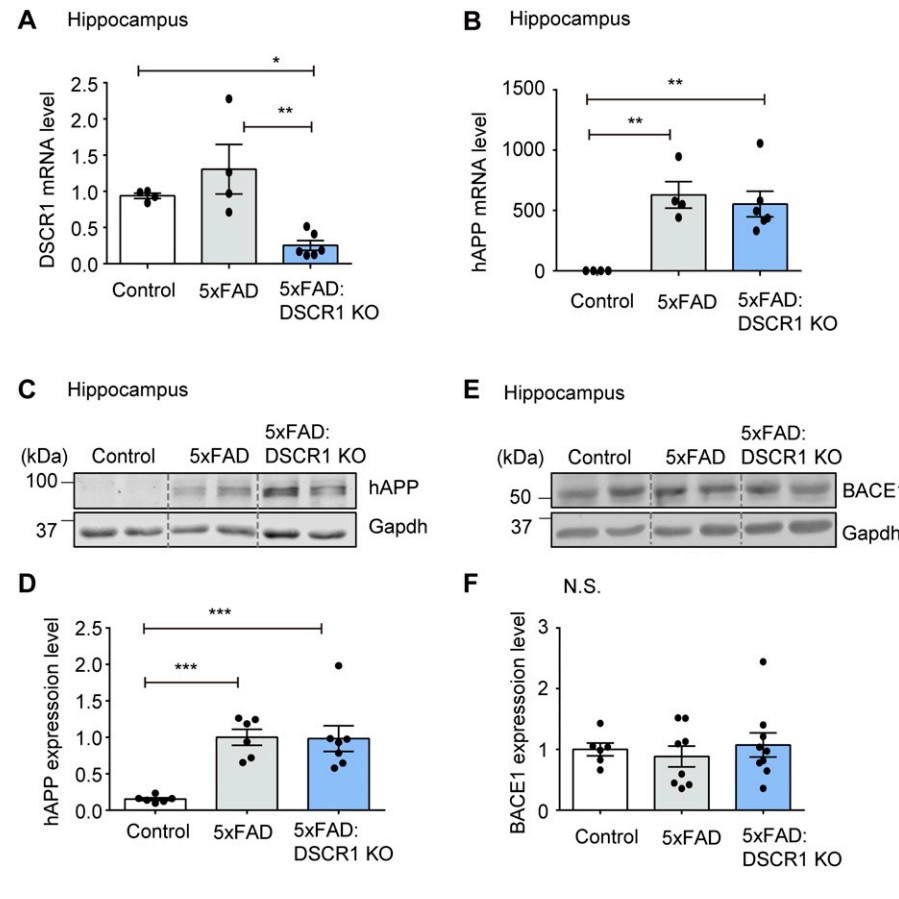

**Figure 2. Amelioration of Aβ deposition in DSCR1 KO/ 5xFAD mice is not mediated by the Aβ synthesis pathway.**
**(A, B)** Relative mRNA expression of DSCR1 and hAPP was measured by qRT-PCR (control, n = 4; 5xFAD, n = 4; DSCR1 KO/5xFAD, n = 6). **(C, D)** Relative protein expression levels of hAPP were measured and compared using Western blots (control, n = 6; 5xFAD, n = 6; DSCR1 KO/5xFAD, n = 7). **(E, F)** Western blot analysis of BACE1 expression and corresponding quantification of relative protein amounts (control, n = 6; 5xFAD, n = 8; DSCR1 KO/5xFAD, n = 9). **(A, B, D, F)** Values are shown as the mean ± SEM and were tested for statistical significance using one-way ANOVA. N.S., not significant, *P < 0.05, **P < 0.01, and ***P < 0.001.
Source data are available for this figure.

lysosomes in DSCR1 KO mice, as seen in Fig 3, we checked whether lysosomal development was enhanced specifically in the microglia. The expression level of LAMP1, a lysosomal marker, was verified in the hippocampus of WT and DSCR1 KO mice to investigate differences in lysosomal development (Eskelinen, 2006). Interestingly, LAMP1 protein levels were increased in the hippocampi of DSCR1 KO mice compared with those in WT mice (Fig 4A and B). We also stained primary microglia with another lysosomal marker, CD68 (Pan et al, 2019), and found that CD68 expression was significantly up-regulated in primary microglia from DSCR1 KO mice (Fig 4C and D).

To confirm whether the up-regulation of CD68 and LAMP1 expression leads to enhanced lysosomal function in DSCR1 KO mice, we assessed the lysosomal degradation ability of DSCR1-deficient microglia. We conducted a lysosomal activity assay that included a self-quenched substrate that emits a FITC fluorescence signal when it is degraded in the lysosome, and measured the fluorescence signal using flow cytometry. As a result, we consistently showed that the microglial lysosomal activity was significantly enhanced in DSCR1-deficient primary microglia compared with WT, as reflected by the increased mean fluorescence intensity of FITC, strongly suggesting that DSCR1 ablation in microglia up-regulated lysosomal degradation activity (Fig 4E and F). Furthermore, we generated a DSCR1-KD BV2 cell line through lentivirus infection, and the knockdown efficiency of DSCR1 was confirmed by qRT-PCR (Fig S4A). We found that DSCR1-KD BV2 cells also showed significantly

enhanced lysosomal activity (Fig S4B and C). Taken together, we suggest that lysosomal activity is up-regulated in DSCR1-deficient microglia and that this could mediate the enhancement of fAβ degradation in microglia of DSCR1 KO/5xFAD mice.

### DSCR1 regulates the microglial transcriptome in BV2 cells

To explore the effect of DSCR1 on the transcriptome of microglia, we performed RNA sequencing (RNA-seq) using scrambled shRNA control and *DSCR1* gene–specific shRNA-expressing BV2 cells treated with or without Alexa Fluor 647–labeled fAβ for 24 h. First, transcriptomic differences between groups were evaluated using correlation analysis, and the heatmap demonstrated that the two biological replicates were highly correlated (Fig 5A). Next, we performed a gene set enrichment analysis and found the genes responsible for IFN-g as indicated by their "responses to interferon-γ," which indicated that they were significantly induced by DSCR1-KD in BV2 cells compared with controls. Moreover, these alterations in gene expression were the most significantly enriched biological processes in DSCR1-KD BV2 cells as a result of treatment with fAβ. (Fig S5).

After the clustering analysis of genes according to their expression patterns, we specifically identified that 904 gene expression levels were increased in the DSCR1-KD BV2 cells and highly up-regulated in the DSCR1-KD BV2 cells stimulated by Aβ (Fig 5B). Interestingly, these transcripts were significantly

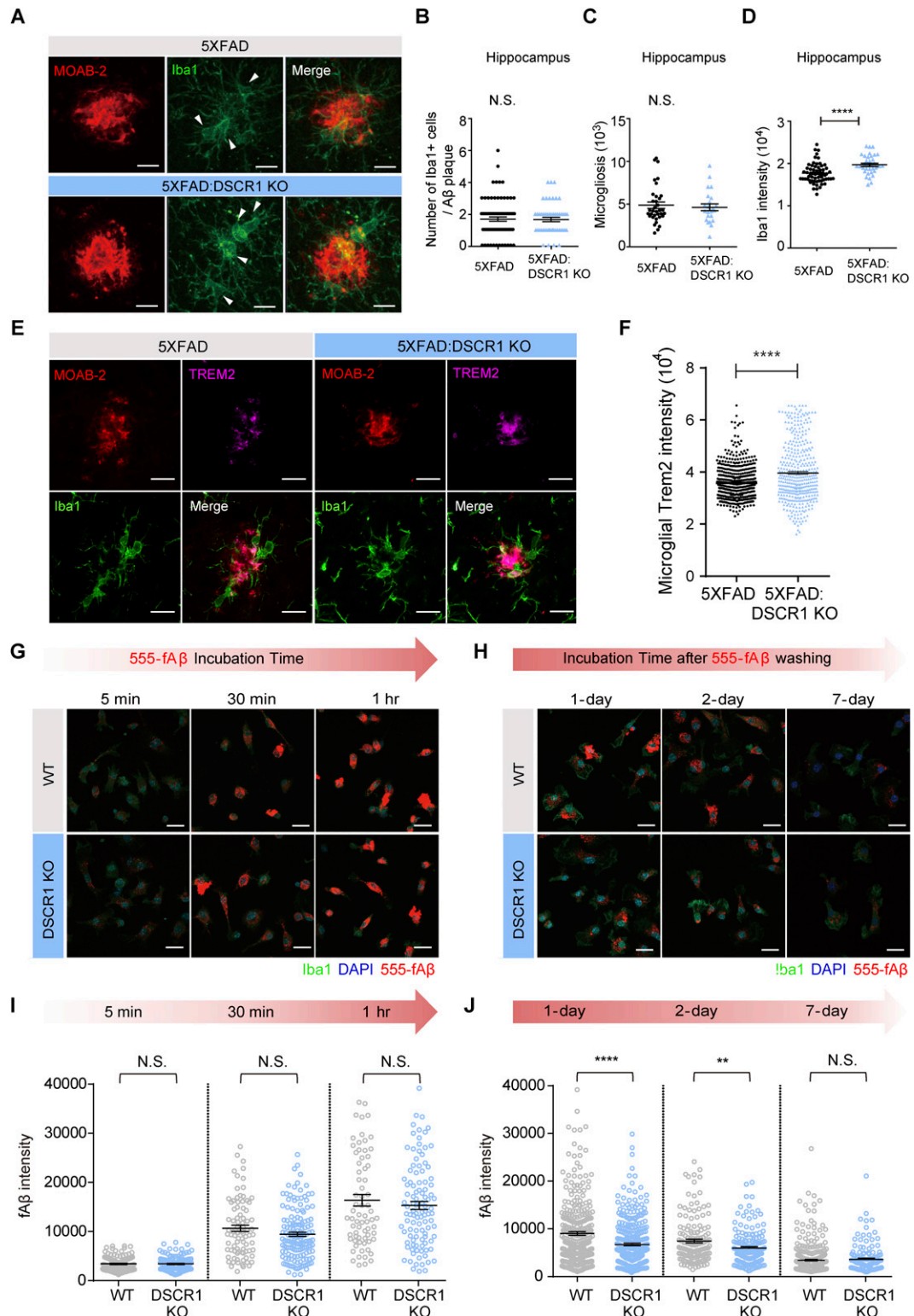

**Figure 3. Plaque-associated microglia from DSCR1 KO/5xFAD mice exhibited increased Iba1 expression, and microglia from DSCR1 KO mice accelerated the degradation of fAβ compared with WT mice.**

**(A)** Representative IHC image of Aβ (MOAB2) and microglia (Iba1) with similar plaque sizes in 5xFAD mice and DSCR1 KO/5xFAD mice. Scale bar = 10 μm. **(B, C, D)** Number of Iba1-positive microglia associated with Aβ plaques (B), microgliosis (C), and Iba1 intensity (D) (5xFAD, n = 78 plaques; DSCR1 KO/5xFAD, n = 50 plaques). **(E)** Representative images of Aβ (MOAB2), TREM2, and microglia (Iba1) in 5xFAD mice and DSCR1 KO/5xFAD mice at 5 mo. Scale bar = 10 μm. **(F)** Quantification of microglial TREM2 intensity (5xFAD, n = 499 Iba1-positive (Iba1+) cells; DSCR1 KO/5xFAD, n = 399 Iba1+ cells). **(G)** Representative images from the engulfment assay. Scale bar = 20 μm. **(H)** Quantification

enriched in various biological processes, such as neutrophil chemotaxis and cellular responses to interferon-γ (Fig 5C and D). Furthermore, knockdown of DSCR1 promoted the up-regulation of inflammatory mediators and members of the IFN-γ signaling pathway, including *CCL20, CCL2, CCL3, INOS, STAT1,* and *JAK1* (Fig 5E), suggesting a functional role of DSCR1 in the regulation of microglia-mediated inflammatory responses.

DSCR1 overexpression in Down syndrome (DS) is known to have multiple roles in the brain, particularly in neurons (Wang et al, 2012; Choi et al, 2019). Notably, early-onset AD is the representative characteristic of individuals with DS (Lott & Head, 2001). The genes for both *APP* and *DSCR1* are located on human chromosome 21 and are overexpressed in DS because of the presence of an extra copy of this chromosome (trisomy 21) (Sun et al, 2011; Wu & Song, 2013; Perluigi et al, 2014). The overexpression of APP, which is cleaved to yield Aβ, has been linked to the increased level of Aβ in the brain; however, DSCR1 is also increased in the postmortem brains of sporadic AD patients (Harris et al, 2007; Wong et al, 2015). Thus, it is conceivable that DSCR1 overexpression could contribute to early-onset AD. Specifically, Aβ and APP are associated with the up-regulation of DSCR1 (Lloret et al, 2011; Wu et al, 2015), and DSCR1 can reciprocally increase Aβ generation (Wang et al, 2014) and the cytotoxicity of Aβ42 aggregates (Lee et al, 2016). Several in vitro studies have shown that DSCR1 overexpression induces additional AD-like pathology, including τ hyperphosphorylation, oxidative stress–induced apoptosis, mitochondrial dysfunction, synaptic abnormality, and neuronal death (Ma et al, 2004; Ermak et al, 2012; Wu & Song, 2013). However, the role of DSCR1 in glial cell populations has not been fully explored. In this study, we discovered a novel effect of DSCR1 on microglia. Interestingly, DSCR1 deficiency in 5xFAD mice improved learning and memory impairment and ameliorated Aβ deposition, which is mediated by enhanced microglial lysosomal activity.

How does the loss of DSCR1 improve microglial activation in AD? Our RNA-seq analysis revealed that the knockdown of DSCR1 induces transcription profiles of inflammation-related factors involved in the IFN-γ signaling pathway in Aβ-stimulated BV2 microglial cells. IFN-γ is known to boost the mTOR-HIF-1a pathway (Cheng et al, 2014) and is down-regulated in the brains of 5xFAD mice (Baruch et al, 2015). In particular, IFN-γ restores microglial phagocytic activity through metabolic reprogramming from oxidative phosphorylation to glycolysis (Baik et al, 2019). These studies suggest that DSCR1 down-regulation may contribute to the activation of microglial function and lead to the improvement of microglial Aβ clearance in AD through modulation of IFN-γ signaling.

DSCR1 contains two major functional domains: the CaN-binding domain and the RNA recognition motif (Mehta et al, 2009). CaN is a serine/threonine phosphatase and a downstream target of DSCR1 (Aubareda et al, 2006). Transcription factor EB (TFEB) is a well-known master regulator of lysosome biogenesis and function (Kim et al, 2021). Interestingly, the downstream gene of TFEB is activated through dephosphorylation by CaN (Sardiello et al, 2009; Medina et al, 2015). As DSCR1 inhibits CaN phosphatase activity, TFEB activity could be increased through the activation and translocation of TFEB by CaN, potentially leading to the up-regulation of lysosome biogenesis and function. Hence, the DSCR1-CaN-TFEB pathway may also be a possible mechanism for up-regulated lysosomal development in DSCR1 KO microglia.

Dysfunction in microglial clearance has emerged as an important risk factor for various neurodegenerative diseases, including AD (Ulland et al, 2017; Hickman et al, 2018). We demonstrated the up-regulation of the engulfment and degradation activity of Aβ in DSCR1-deficient mice through microglia-specific in vitro experiments. Furthermore, loss of DSCR1 in 5xFAD mice, a model of AD, ameliorated memory loss and Aβ deposition and increased microglial activation. Therefore, our data suggest that the regulation of DSCR1 could be a potential therapeutic target for microglial Aβ clearance in AD.

# Materials and Methods

### Animal experimental model

Five familial AD (5xFAD) transgenic mice (34840; Jackson Laboratory) were obtained from Dr. Inhee Mook-Jung at Seoul National University. 5xFAD mice expressing hAPP and *PSEN1* transgenes with five AD-related mutations were maintained by backcrossing transgenic animals with a C57BL/6J/SJL hybrid at every generation. DSCR1 KO mice were obtained from K Baek at Sungkyunkwan University (Ryeom et al, 2003), and their genetic background was C57BL/6J (B6). The 5xFAD mice were crossed with DSCR1 KO mice to generate DSCR1 KO/5xFAD hybrid mice. To match their genetic backgrounds, 5xFAD mice were crossed with B6 mice. 5-mo-old male 5xFAD and DSCR1 KO/5xFAD mice and 3-mo-old male WT and DSCR1 KO mice were used for the experiment. All mice were tested using protocols approved by the Institutional Animal Care and Use Committee of the Ulsan National Institute of Science and Technology.

### Immunohistochemistry

Mice were anesthetized with tribromoethanol (Avertin) by intraperitoneal injection and perfused with ice-cold 0.1 M PBS. The skull was removed using scissors to isolate the brain tissue. Brain hemispheres were fixed with 4% PFA at 4°C overnight and postfixed for 2 d by soaking them in 30% sucrose for cryoprotection. After fixation and cryoprotection, the brain was embedded in a mold with tissue-freezing medium and rapidly frozen in a deep freezer. Frozen brains were sectioned as 40-μm-thick slices using a cryostat (Leica). Brain sections were washed three times with PBS for 10 min and incubated with 88% formic acid for 20 min at RT to enhance MOAB2

---

of internalized fluorescence fAβ intensity in both cell types at different time points after fAβ treatment (WT, n = 71–124 cells; DSCR1 KO, n = 105–144 cells). **(I)** Representative images from the degradation assay. Scale bar = 20 μm. **(J)** Quantification of degraded fAβ intensity at different time points after fAβ washing (WT, n = 153–333 cells; DSCR1 KO, n = 135–351 cells). **(B, C, D, F, I, J)** Values are shown as the mean ± SEM and were tested for statistical significance using a two-tailed unpaired *t* test. N.S., not significant, **P < 0.01, and ****P < 0.0001.

immunoreactivity by dissolving amyloid fibrils. Brain sections were then rinsed with PBS three times and incubated with sodium citrate buffer (10 mM, pH 6) for 30 min at 75°C. The sections were then cooled to RT and washed again three times with PBS. Next, the brain sections were incubated with blocking buffer and 5% normal horse serum in PBS-T (0.1 M PBS and 0.3% Triton X-100) for 1 h. Finally, sections were incubated with MOAB2 (1:500, NBP2-13075; Novus), TREM2 (1: 100, MAB17291-100; R&D Systems), and Iba1 (1:500, 019-19741, 011-27991; Wako) primary antibodies with 2% normal horse serum in PBS-T at 4°C overnight. After 1-d incubation, brain sections were washed three times with PBS-T and then incubated with donkey Alexa Fluor 568 anti-mouse IgG (1:500; Thermo Fisher Scientific) and donkey Alexa Fluor 488 anti-rabbit IgG (1:500; Thermo Fisher Scientific) secondary antibodies in PBS-T for 2 h. After washing with PBS and DAPI staining (1:1,000; Thermo Fisher Scientific) for 10 min, the brain sections were washed three times with PBS-T and mounted on a glass slide with mounting solution (Invitrogen). For Thioflavin-S staining, free-floating brain sections were dehydrated by a graded series of ethanol (70% and 80%; 1 min each). The sections were incubated in 1% Thioflavin-S (in 80% ethanol) for 15 min and then washed with a series of ethanol (80% and 70%; 1 min each). The sections were then washed twice with water for 5 min and washed one final time in 1× PBS (1 × 10 min).

## MWM

The 5-mo-old 5xFAD and 5xFAD/DSCR1 KO mice were tested with the MWM according to a previous study (Nunez, 2008). Each mouse was trained for five consecutive days to memorize the location of the hidden platform, and their escape latency was measured. The probe trial without the platform was tested for 30 s on the last training day. Latency to the platform, the percentage of total time spent in each platform quadrant, and platform crossing events were recorded and analyzed using the Smart Video Tracking software v3.0 (Harvard apparatus).

## Total RNA sampling and qRT-PCR

The hippocampi were isolated from the brain hemispheres and lysed with TRIzol reagent (10296028; Invitrogen) to extract the total RNA. cDNA was prepared from the total RNA using a high-capacity RNA-to-cDNA kit (4388950; Life Technologies). The mRNA expression levels were measured by qRT-PCR using SYBR Green (RT500; Enzynomics). Primers used for qRT-PCR are listed in Table 1.

## RNA-seq

RNA-seq was performed by Novogene using total RNA samples of shscramble and shDSCR1 BV2 cells with or without Aβ treatment (n = 2 per group). Gene expression patterns were analyzed with the normalized FPKM value of each gene and clustered using the K-means clustering algorithm with the "ClusterR" R package (version 1.2.5) for genes with similar expression patterns. Pathway enrichment analysis was performed using DAVID (https://david.ncifcrf.gov/), and the heatmap of genes related to neutrophil chemotaxis and cellular responses to interferon-γ was visualized using the "pheatmap" R package (version 1.0.12).

**Table 1.  Forward and reverse primers used in the qRT-PCR experiments.**

| Oligo name | Oligo sequence (5′ → 3′) |
|---|---|
| F: β-actin | AGCCATGTACGTAGCCATCC |
| R: β-actin | CTCTCAGCTGTGGTGGTGAA |
| F: DSCR1_Exon5 | TGCACAAGACCGAGTTCCTGG |
| R: DSCR1_Exon6 | TGTTTGTCGGGATTGGGCGG |
| F: hAPP | CATTGGACTCATGGTGGGCG |
| R: hAPP | CTGCATCTTGGACAGGTGGC |
| F: Stat1 | GCCTCTCATTGTCACCGAAGAAC |
| R: Stat1 | TGGCTGACGTTGGAGATCACCA |
| F: Jak1 | CTGTCTACTCCATGAGCCAGCT |
| R: Jak1 | CCTCATCCTTGTAGTCCAGCAG |
| F: Ccl20 | GTGGGTTTCACAAGACAGATGGC |
| R: Ccl20 | CCAGTTCTGCTTTGGATCAGCG |
| F: Ccl2 | GCTACAAGAGGATCACCAGCAG |
| R: Ccl2 | GTCTGGACCCATTCCTTCTTG G |
| F: Ccl3 | TGTACCATGACACTCTGCAAC |
| R: Ccl3 | CAACGATGAATTGGCGTGGAA |
| F: iNOS | AGCACAGGAAATGTTTCAGC |
| R: iNOS | AAGTCATGTTTGCCGTCACT |

## Western blot

The hippocampus was isolated from the brain hemisphere and lysed in NETN lysis buffer containing protease inhibitor PMSF. Equal amounts of the lysates were loaded and separated using SDS–PAGE. Next, the proteins were transferred to nitrocellulose membranes (GE Healthcare) and blocked using 5% skim milk in TBST for 1 h. The membrane was incubated with indicative primary antibodies at 4°C overnight: GAPDH (1:1,000; Cell Signaling Technology), hAPP(1:1,000; Invitrogen), LAMP1 (1: 1,000; Santa Cruz), and BACE1 (1:500; Santa Cruz). The membranes were then incubated with IRDye-conjugated anti-mouse or rabbit secondary antibodies for 2 h at RT. Proteins were detected using Odyssey CLx and quantified using the ImageJ software.

## Primary microglial culture

Male and female P0-2 pups were used for primary microglial culture. The brain was isolated from the skull, and meninges were removed using forceps in 1× HBSS (14065-056; Gibco). The cerebral cortex and hippocampus were transferred to Ca²⁺- and Mg²⁺-free 1× HBSS and cut into small pieces. The pieces were then transferred into pre-warmed trypsin and incubated at 37°C for 30 min. Undigested tissue was removed using a 70-μm cell strainer, and the brain cell mixture was centrifuged. The pull-down cells were seeded into a PDL-coated 225T flask with 10% FBS, 10% HS, and 1% P/S in DMEM and incubated at 37°C in water-saturated 5% CO₂ and 95% air atmosphere for 2 wk. After the 2-wk incubation, primary microglia were isolated by

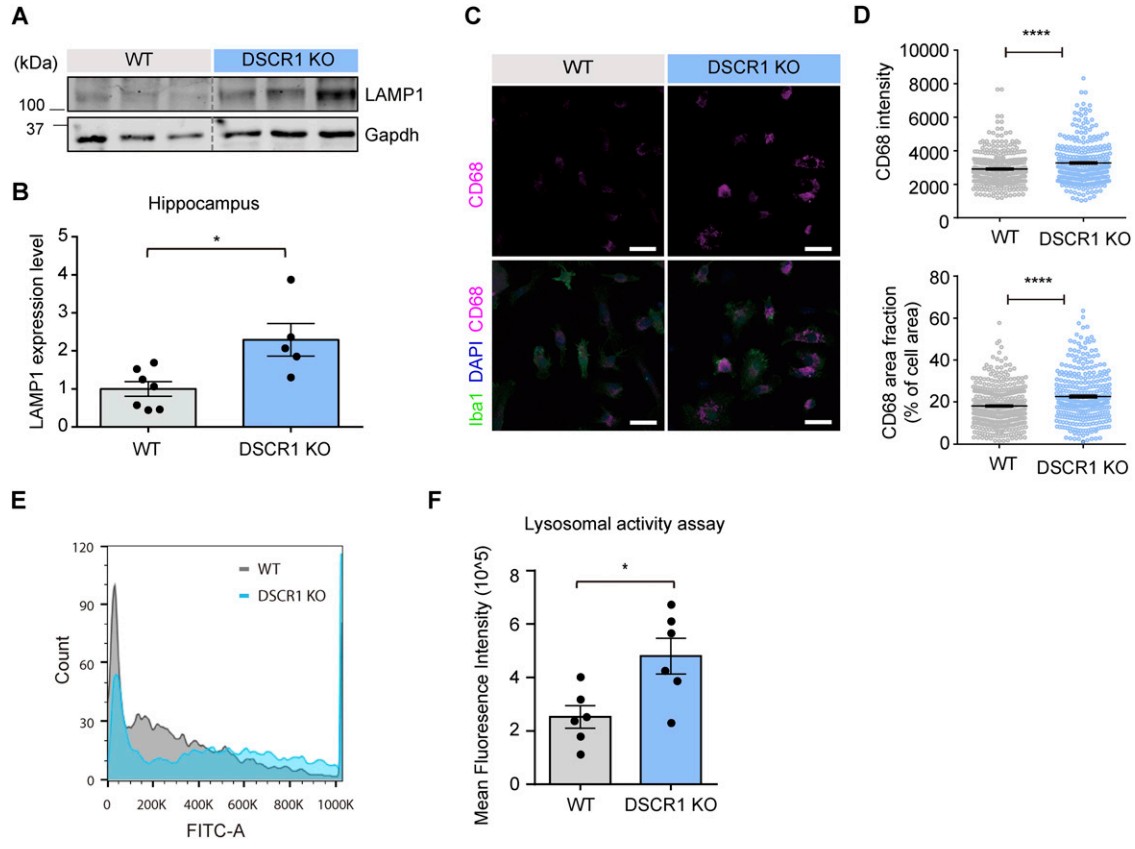

**Figure 4. Lysosomal development was improved, and lysosomal activity was enhanced in microglia from DSCR1 KO mice.**
**(A, B)** Relative protein expression of LAMP1 was measured using Western blots (WT, n = 7; DSCR1 KO, n = 5). **(C)** Representative images of the CD68⁺ immunostaining from the immunocytochemistry analysis of the primary microglia of WT mice and DSCR1 KO mice. Scale bar = 20 $\mu$m. **(D)** CD68 intensity and area fraction values were quantified using primary microglia from both mice (WT, n = 340 cells; DSCR1 KO, n = 308 cells). **(E)** Histogram of FITC fluorescence in microglia prepared from WT and DSCR1 KO mouse brains (n = 5,000 cells). **(F)** Quantification of mean fluorescence intensity using FITC fluorescence in primary microglia from WT and DSCR1 KO mice (n = 6 independent experiments). **(B, D)** Values are shown as the mean ± SEM and were tested for statistical significance using a two-tailed unpaired $t$ test (B) and a paired $t$ test (D). *$P < 0.05$ and ****$P < 0.0001$. **(F)** Values are shown as the mean ± SEM and were tested for statistical significance using a two-tailed unpaired $t$ test. *$P < 0.05$.
Source data are available for this figure.

shaking the flask at 35$g$ for 4 h, and isolated primary microglial cells were seeded into a PDL-coated culture dish with a coverslip for the experiment.

### fA$\beta$ engulfment and degradation assay

HiLyte Fluor 555–labeled $\beta$-amyloid 1-42 (AnaSpec) was used to prepare 555-fA$\beta$ as described previously (Baik et al, 2019). For the engulfment assay, 1 555-fA$\beta$ was added to the primary microglial culture media from WT and DSCR1 KO mice. The primary microglial cells were incubated for 5 min, 30 min, and 1 h at 37°C, washed with 1× PBS, and then fixed with 4% PFA for 20 min. For the degradation assay, 1 $\mu$M of 555-fA$\beta$ was added to the primary microglial culture medium for 1 h at 37°C; the primary microglial cells were then washed with 1× PBS twice and incubated with 555-fA$\beta$–free culture medium for one, two, and seven additional days (Paresce et al, 1997). The primary microglial cells were also washed with 1× PBS and fixed with 4% PFA for 20 min. The PFA-fixed primary microglial cells were stained with an Iba1 antibody to identify microglia through

immunocytochemistry for both engulfment and degradation assays.

### Immunocytochemistry

BV2 cells and primary microglial cells were seeded into uncoated glass coverslips in 24-well culture plates. PFA-fixed cells were washed with PBS and incubated with permeabilization buffer (0.1 M PBS and 0.2% Triton X-100) for 15 min at RT. Then, cells were incubated with blocking buffer (0.1 M PBS, 0.2% Triton X-100, and 1% BSA) for 1 h at RT and treated with Iba1 (1:500, 019-19741; Wako), DSCR1 (1:100, RP3941; ECM Biosciences), and CD68 (1:200, ab53444; Abcam) primary antibodies in blocking buffer at 4°C overnight. The primary antibodies were washed with PBS, and then, the cells were immersed in Alexa Fluor 488–conjugated anti-rabbit IgG (1:500; Invitrogen) or Alexa Fluor 647–conjugated anti-rat IgG (1:500; Invitrogen) secondary antibodies for 2 h at RT. After washing the secondary antibodies with PBS, coverslips containing seeded cells were mounted on glass slides with mounting solution (Invitrogen).

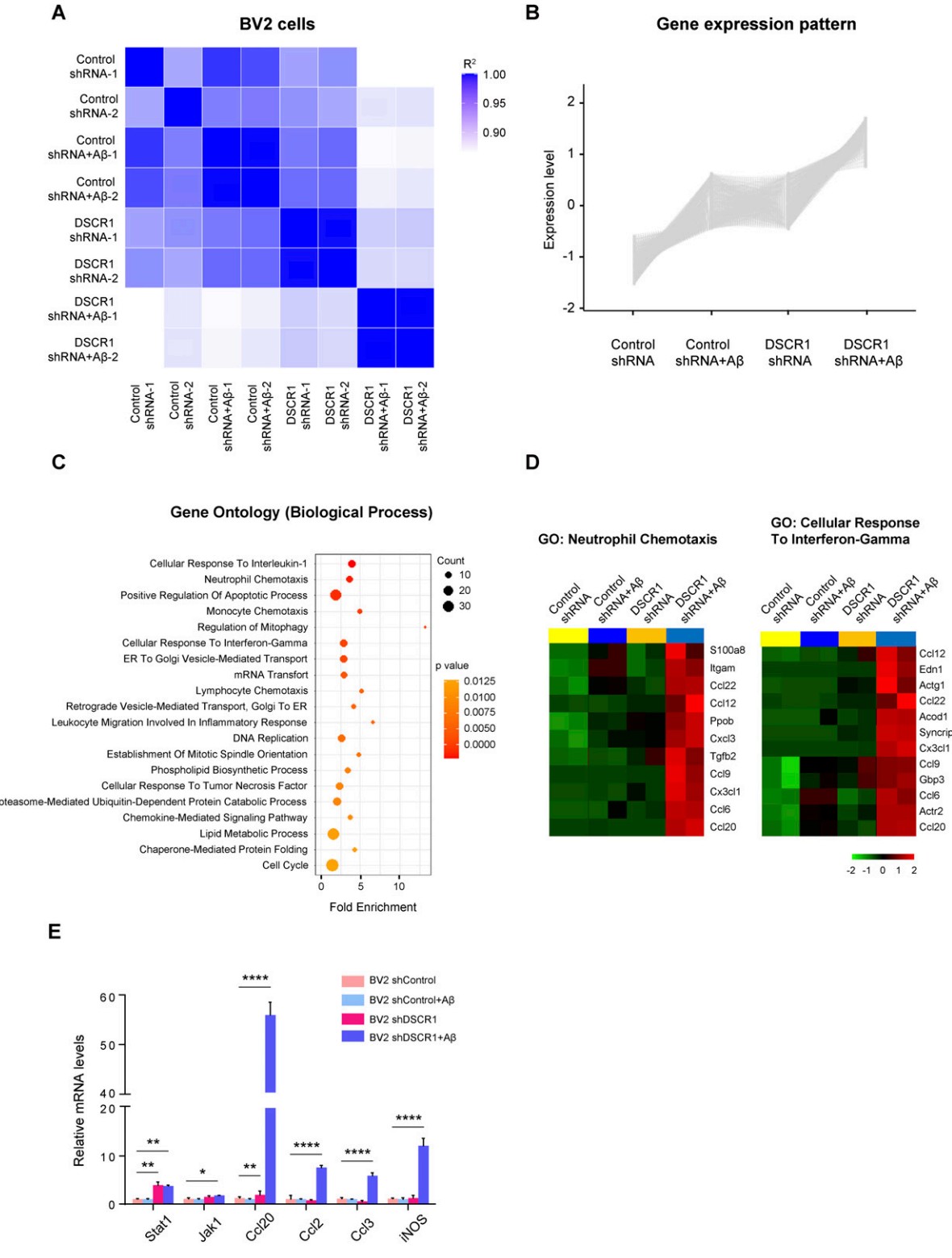

**Figure 5. Knockdown of DSCR1 elicits transcriptional changes in BV2 cells with Aβ plaques.**
**(A)** Pearson's correlation heatmap showing a strong correlation between RNA-seq samples from control and DSCR1-knockdown BV2 cells with or without Aβ treatment. **(B)** Gene expression pattern based on the clustering analysis. **(C)** Top 20 gene ontology biological processes identified after the pathway enrichment analysis of the obtained cluster. **(D)** Heatmap representing the gene expression levels of the selected biological processes among the top 20 biological processes. **(E)** Relative mRNA expression of the inflammatory cytokines and IFN-γ–induced genes (n = 3 per group). **(E)** Values are shown as the mean ± SEM and were tested for statistical significance using one-way ANOVA. *$P < 0.05$, **$P < 0.01$, and ****$P < 0.0001$.

### Lentivirus production and infection

DSCR1 shRNA (shDSCR1) was cloned into the pLKO.1-TRC vector according to the manufacturer's protocol (http://www.addgene.org/protocols/plko/). Lentivirus was produced in HEK293T cells by co-transfection with the pLKO.1-TRC vector (shscramble or shDSCR1), pCMV-dR8.2 dvpr, and pCMV-VSV-G. BV2 cells were treated with the supernatant containing the lentivirus (shscramble or shDSCR1) along with poly-l-lysine and polybrene for 48 h. After lentivirus incubation, the infected BV2 cells were selected using puromycin (30 $\mu$g/ml) for 72 h. A single colony was isolated from the puromycin-treated BV2 cells, and knockdown efficiency was determined by qRT-PCR. The shscramble and shDSCR1 sequences are listed in Table 2.

### Lysosomal activity assay

Lysosomal activity assay was performed as described in the manufacturer's manual (ab234622; Abcam). Lysosomal activity was visualized by FITC fluorescence from a self-quenched substrate. After 1-h incubation with the self-quenched substrate, microglia prepared from WT and DSCR1 KO mouse brains and BV2 cells were fixed with 4% PFA and 0.1% saponin in PBS for 20 min and washed with 0.5% Tween-20 and 1% BSA in PBS. The cell suspension was filtered using a 70-$\mu$m cell strainer (Falcon) to purify single cells for FACS analysis. The FITC fluorescence signal of each cell was measured by FACS (BD LSRFortessa), and the histogram of each cell population was merged using the FlowJo software.

### In vivo BBB permeability assay

To estimate dextran–tetramethylrhodamine leakage, we injected 2 mg/20 g of body weight fluorescence-tagged dextran (D1817; Thermo Fisher Scientific) into the left ventricle and allowed it to cross the intact BBB for 10 min. After perfusion, brains were isolated, fixed, and immersed into a frozen section medium, then treated as described for immunohistochemistry. The presence of fluorescence in the cortical and striatum areas was focused and calculated to quantify BBB leakage.

### Cycloheximide (CHX) chase assay

Mice were intraperitoneally injected with 160 mg/kg CHX and euthanized at various time points after the CHX injection. After perfusion, the hippocampus was isolated from the brain and then treated as described for Western blot analysis.

### Image analysis

Immunofluorescence-stained brain tissues and primary microglial cells were imaged with an LSM 780 microscope using the ZEN software. The maximum intensity projection method was applied to each Z-stack image for quantification. The A$\beta$ staining area (%) was calculated as the MOAB2 staining area fraction by the total DAPI-stained area using the ImageJ software. The size and number of A$\beta$ plaques were calculated and counted using the particle analysis tool in ImageJ. For the fA$\beta$ engulfment and degradation assay, the 555-fA$\beta$ mean intensity was measured within a single Iba1-positive area to quantify only microglia-enclosed fA$\beta$ using ImageJ (Pan et al, 2019).

### Statistical analysis

Values are shown as the mean ± SEM. Statistical significance was determined using unpaired $t$ tests, paired $t$ tests, one-way ANOVA followed by Tukey's test, and repeated-measures two-way ANOVA. Statistical significance was analyzed using the GraphPad Prism 6.01 software.

### Ethical approval

All animals were maintained and used following the protocol (UNISTIACUC-21-17) approved by the Institutional Animal Care and Use Committee of the Ulsan National Institute of Science and Technology.

## Data Availability

Data sets generated from the current study are available from the corresponding author.

## Supplementary Information

## Acknowledgements

We thank Dr. Inhee Mook-Jung of Seoul National University for kindly gifting the 5xFAD mice. This work was supported by grants from the Korea Health Technology R&D Project through the Korea Health Industry Development Institute (KHIDI; HI14C1277), and the Basic Science Research Program to J Park (NRF-2018R1A5A1024340 and NRF-2021R1A2C2005499), M Kim (NRF-2020R1A2C4001503), and C-S Kim (NRF-2022R1I1A1A01069481) through the National Research Foundation.

### Author Contributions

C Choi: conceptualization, data curation, formal analysis, validation, investigation, and writing—original draft, review, and editing.

### Table 2. shscramble and shDSCR1 sequences.

| shRNA name | shRNA sequences (5′ -> 3′) |
|---|---|
| shscramble | CCTAAGGTTAAGTCGCCC TCGCTCGAGCGAGGGCGACTTAACCTTAGG |
| shDSCR1 | CCGTCATAAATTACGATCTTT CTCGAGAAAGATCGTAATTTATGACGG |

H Kim: conceptualization, data curation, formal analysis, investigation, and writing—original draft.

J Oh: data curation, formal analysis, investigation, and methodology.

C Park: data curation, software, formal analysis, and validation.

M Kim: data curation, funding acquisition, investigation, and methodology.

C-S Kim: formal analysis, funding acquisition, and investigation.

J Park: conceptualization, resources, data curation, formal analysis, supervision, funding acquisition, validation, investigation, methodology, and writing—original draft, review, and editing.

## Conflict of Interest Statement

The authors declare that they have no conflict of interest.

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
