## [Reviewer comments · Life Science Alliance]

Life Science Alliance

DSCR1 deficiency ameliorates the A β pathology of Alzheimer's disease by enhancing microglial activity

Chiyeol Choi, Hyerin Kim, Jiyoung Oh, Chanho Park, Min Kim, Chu-Sook Kim, and Jiyoung Park

DOI: <https://doi.org/10.26508/lsa.202201556>

Corresponding author(s): Jiyoung Park, Ulsan National Institute of Science and Technology

Review Timeline:

Submission Date:	2022-06-09
Editorial Decision:	2022-07-26
Revision Received:	2022-11-14
Editorial Decision:	2022-11-16
Revision Received:	2022-11-17
Accepted:	2022-11-21

Transaction Report:

July 26, 2022

Re: Life Science Alliance manuscript #LSA-2022-01556-T

Prof. Jiyoung Park
Ulsan National Institute of Science and Technology
Biological Sciences
50 UNIST-gil
Ulsan, NA 44919
KOREA, REPUBLIC OF

Dear Dr. Park,

Thank you for submitting your manuscript entitled "DSCR1 deficiency ameliorates the A β pathology of Alzheimer's disease by enhancing microglial activity" to Life Science Alliance. The manuscript was assessed by expert reviewers, whose comments are appended to this letter. We invite you to submit a revised manuscript addressing the Reviewer comments.

Thank you for this interesting contribution to Life Science Alliance. We are looking forward to receiving your revised manuscript.

Sincerely,

B. MANUSCRIPT ORGANIZATION AND FORMATTING:

Reviewer #1 (Comments to the Authors (Required)):

"DSCR1 deficiency ameliorates the A β pathology of Alzheimer's disease by enhancing microglial activity" by Kim et al investigated the functional role of DSCR1 in the AD pathogenesis from a beta amyloidosis model. By combining the analysis of germline knockout from 5XFAD mice and characterization of gene knockdown BV-2 microglia cell line in vitro, the authors revealed a pathogenic effect of DSCR1 and eluded to a role of DSCR1 in regulating microglial activities. The study design was straightforward, and experiments were carried out reasonably. However, several major conceptual gaps prevent a cohesive comprehension of the involvement of DSCR1 in AD.

1. Authors did not provide any definitive evidence to substantiate DSCR1 protein expression in microglia in AD brain. This is a critical miss as this report centers on the primary function of the gene in microglia. How is it regulated by glia cells surrounding the plaques? What is the subcellular location of this molecule?
2. DSCR1 is clearly associated with Down Syndrome, where its gene dose escalates. Despite the phenotype of insufficiency of DSCR1, evidence is lacking to demonstrate a pathogenic impact of DSCR1 at higher doses. Many genes are required for the proper development and maturation of the brain. A phenotype in knockout may not immediately translate into a gain of function in disease.
3. Despite having access to DSCR1^{-/-} microglia, the authors studied the function of DSCR1 solely from KD BV-2 cells. BV-2 cells have limited relevance to primary microglia in their molecular configurations. Additionally, modeling microglia activation with A β peptide incubation in vitro can not recapitulate the complex interactions between plaques and microglia in vivo. Therefore, an analysis of DSCR1^{-/-} microglia needs to be performed to validate the main claims of BV-2 experiments.
4. The authors stressed that IFN- γ the response was the most functionally impacted by DSCR1 in microglia. This is very perplexing, as Fig 5C clearly showed that Cellular Response to Interleukine-1 was most profoundly affected (by fold change), and so was Regulation of Apoptotic Process (by counts), both of which were no less significant than that to IFN- γ . Since we know both signaling pathways critically affect microglia, the reason why IFN- γ response was pursued in a biased way needs explanation.

Minor points:

- Characterization of microglia is rather simplistic and incomplete. Besides analyzing Iba1, other DAM microglia markers, such as Trem2 or Clec7a should be examined to substantiate the main conclusion.
- Despite the claim that DSCR1^{-/-}5xFAD brain contains significantly reduced amyloid plaques (Fig 1), it does not seem that the plaques are smaller (Fig 3a). Is it so? The authors need to quantify and clarify this point

Reviewer #2 (Comments to the Authors (Required)):

This report provides data supporting DSCR1 (RCAN1) as a potential therapeutic target in AD. The authors showed that KO of the gene substantially decreased plaque load in the 5XFAD amyloid transgenic. The suggest that his is due to increased uptake and processing by microglia through activation of the endocytic-lysosomal system based on lack of change iAPP mRNA level,, increased Iba1 and LAMP1, CD68 staining and improved lysosomal protein expression in BV2 cells with shRNA knockdown. The authors propose that DSCR1 is a critical mediator of microglial degradation of amyloid plaques and a new potential microglial therapeutic target.

The data on the KO are novel and the effect on plaque load is dramatic and well worth reporting. However, the strength of the evidence attributing this solely to microglial function changes remains too limited to support the specific claims made in the results:

1. While APP mRNA has been assessed, there is not direct data presented on synthetic rates. The author's paper rests on demonstration that breakdown/clearance is independently responsible. I urge direct measures of synthesis, e.g., with 35Smet pulse-chase (e.g., Savage, J Neurosci, 1998).
2. Attribution of all of the relevant effects to microglia is unwarranted. DSCR1 is widely expressed. Other hypotheses are suggested, e.g., could enhanced clearance occur at the BBB with changes in endothelial cell function?
3. The evidence for microglial activation for phagocytosis seems incomplete. Markers of activation were limited. Data concerning DAM gene (Keren-Schaul, Cell, 2017) expression in the microglia would add meaningfully to the case.
4. Finally, as minor details of interest, can the authors augment their descriptions of plaque morphology? E.g., are dense core plaques increased?

Thank you for the opportunity to submit a revised draft of our manuscript to *Life Science Alliance*. We thank the Editors and Reviewers for their positive and constructive comments that contributed towards the considerable improvement of the manuscript. We have revised the manuscript based on the suggestions of the editor and reviewers.

Point-by-point response to the reviewers:

Reviewer #1 (Comments to the Authors (Required)):

"DSCR1 deficiency ameliorates the A β pathology of Alzheimer's disease by enhancing microglial activity" by Kim et al investigated the functional role of DSCR1 in the AD pathogenesis from a beta amyloidosis model. By combining the analysis of germline knockout from 5XFAD mice and characterization of gene knockdown BV-2 microglia cell line in vitro, the authors revealed a pathogenic effect of DSCR1 and eluded to a role of DSCR1 in regulating microglial activities. The study design was straightforward, and experiments were carried out reasonably. However, several major conceptual gaps prevent a cohesive comprehension of the involvement of DSCR1 in AD.

We thank Reviewer 1 for their positive and insightful comments for the improvement of the manuscript.

1. Authors did not provide any definitive evidence to substantiate DSCR1 protein expression in microglia in AD brain. This is a critical miss as this report centers on the primary function of the gene in microglia. How is it regulated by glia cells surrounding the plaques? What is the subcellular location of this molecule?

*We thank the reviewer to point this out. We have confirmed that DSCR1 was mainly distributed in the whole cell area of primary microglia, and protein expression of DSCR1 was markedly downregulated by fA β treatment for 24h (**Reviewer Fig. 1**). **This new data is now added into *Supplementary Fig 3*. *Figure legends and main text are also accordingly modified (Supplementary p.3, line 1-9 and p.8, line 13-15).****

Reviewer Figure 1. DSCR1 expression was reduced by the treatment of fA β in primary microglial cells. (A) Representative images of immunofluorescence staining against DSCR1, Iba1, and DAPI with or without HiLyte Fluor 555-labeled fA β . The white box area is magnified in the right panels. Scale bar = 100 μ m in the large panel and 20 μ m

in the magnified images. (B) Quantification of microglial DSCR1 intensity (Primary microglia, n = 345 Iba1-positive (Iba1+) cells; Primary microglia treated with fA β , n = 267 Iba1+ cells). (C) The area of DSCR1 expression was quantified in fA β -treated primary microglia compared to the control cells. (n = 3 independent experiments). **(B and C)** Values are shown as mean \pm SEM and tested for statistical significance using a two-tailed unpaired t-test. *P < 0.05.

2. DSCR1 is clearly associated with Down Syndrome, where its gene dose escalates. Despite the phenotype of insufficiency of DSCR1, evidence is lacking to demonstrate a pathogenic impact of DSCR1 at higher doses. Many genes are required for the proper development and maturation of the brain. A phenotype in knockout may not immediately translate into a gain of function in disease.

As the reviewer pointed out, the effect of overexpression of DSCR1 in Alzheimer's disease is supported by various studies. Early onset Alzheimer's disease is the representative characteristic of individuals with Down syndrome (DS) (1). Both amyloid precursor protein (APP) and DSCR1 are located on human Chromosome 21 (HSA21) and overexpressed in DS due to an extra copy (trisomy 21) (2-4). People speculated overexpression of APP, which is cleaved to yield A β , has been linked to the increased level of amyloid beta in the brain, but DSCR1 is also increased in the postmortem brains of sporadic AD patients (5,6). Thus, it is conceivable that DSCR1 overexpression could contribute to early-onset Alzheimer's disease (AD). Particularly, amyloid-beta and APP are linked upregulation of DSCR1 (7, 8), and DSCR1 can reciprocally increases A β generation (9) and cytotoxicity of A β 42 aggregates (10). Several in vitro studies have shown that DSCR1 overexpression induces additional AD-like pathology, including tau hyperphosphorylation, oxidative stress-induced apoptosis, mitochondrial dysfunction, synaptic abnormality, and neuronal death (4, 11, 12). **Now, we added this issue in the discussion (Please check out the revised discussion, p.11, line 21-23 and p.12, line 1-12).**

Our study of DSCR1 deficiency in microglia provides an opportunity for potential drug targets of AD, but the clinical interpretation of DSCR1 knockout would be complicated. Therefore, we suggest that this question, while clearly fascinating, is beyond the scope of this paper and should be addressed in the future works.

1. Down syndrome and Alzheimer's disease: a link between development and aging. *Ment Retard Dev Disabil Res Rev.* 2001;7:172–8.
2. Neuropathological role of PI3K/Akt/mTOR axis in Down syndrome brain. *Bioch Biophys Acta (BBA) - Mol Basis Dis.* 2014;1842:1144–53.
3. Regulator of calcineurin 1 (RCAN1) facilitates neuronal apoptosis through caspase-3 activation. *J Biol Chem.* 2011;286:9049–62.
4. Regulation of RCAN1 translation and its role in oxidative stress-induced apoptosis. *FASEB J.* 2013;27:208–21.
5. RCAN1-1L is overexpressed in neurons of Alzheimer's disease patients. *FEBS J.* 2007;274:1715–24.
6. RCAN1 overexpression promotes age-dependent mitochondrial dysregulation related to neurodegeneration in Alzheimer's disease. *Acta Neuropathol.* 2015;130:829–43.
7. Amyloid- β toxicity and tau hyperphosphorylation are linked via RCAN1 in Alzheimer's disease. *JAD.* 2011;27:701–9.
8. Amyloid- β precursor protein facilitates the regulator of calcineurin 1-mediated apoptosis by downregulating proteasome subunit α type-5 and proteasome subunit β type-7. *Neurobiol Aging.* 2015;36:169–77.
9. RCAN1 increases A β generation by promoting N-glycosylation via oligosaccharyltransferase. *Curr Alzheimer Res.* 2014;11:332–9.
10. The calcineurin inhibitor Sarah (Nebula) exacerbates A β 42 phenotypes in a *Drosophila* model of Alzheimer's disease. *Dis Model Mech.* 2016;9:295–306.
11. Chronic expression of RCAN1-1L protein induces mitochondrial autophagy and metabolic shift from oxidative phosphorylation to glycolysis in neuronal cells. *J Biol Chem.* 2012;287:14088–98.
12. Aggregate formation and synaptic abnormality induced by DSCR1. *J Neurochem.* 2004;88:1485–96.

3. Despite having access to DSCR1^{-/-} microglia, the authors studied the function of DSCR1 solely from KD BV-2 cells. BV-2 cells have limited relevance to primary microglia in their molecular configurations. Additionally, modeling microglia activation with Abeta peptide incubation in vitro can not recapitulate the complex interactions between plaques and microglia in vivo. Therefore, an analysis of DSCR1^{-/-} microglia needs to be performed to validate the main claims of BV-2 experiments.

We thank the reviewer for pointing this out, and the reviewer's concerns are well acknowledged. Following the reviewer's suggestion, we performed a new experiment with primary microglia isolated from DSCR1 KO mice as compared to Wild-type mice. In this new analysis, we consistently showed that the microglial lysosomal activity was significantly enhanced in DSCR1-deficient primary microglia compared to WT, as reflected by increased MFI of FITC, strongly suggesting that DSCR1 ablation in microglia upregulated lysosomal degradation activity (**Reviewer Fig. 2**). **This new data is now added into Revised Fig 4E, F, and original Fig 4E-G is replaced by Supplementary Fig S4. Figure legends and main text are also accordingly modified (p.33, line 1-10, p.10, line 9-15, Supplementary p.4, line 1-6 and p.10, line 17-18).**

Reviewer Figure 2. (A) Histogram of FITC fluorescence in microglia isolated from brain of DSCR1 KO mice compared to WT (n = 5000 cells). (B) Quantification of mean fluorescence intensity with the FITC in primary microglia from WT and DSCR1 KO mice (n = 6 independent experiments). (B) Values are shown as mean ± SEM and tested for statistical significance using a two-tailed unpaired t-test. *P < 0.05.

4. The authors stressed that IFN- γ the response was the most functionally impacted by DSCR1 in microglia. This is very perplexing, as Fig 5C clearly showed that Cellular Response to Interleukine-1 was most profoundly affected (by fold change), and so was Regulation of Apoptotic Process (by counts), both of which were no less significant than that to IFN- γ . Since we know both signaling pathways critically affect microglia, the reason why IFN- γ response was pursued in a biased way needs explanation.

We thank the reviewer for pointing this out. We fully agree with the reviewer that both Cellular Response to Interleukine-1 (by fold change) and Regulation of Apoptotic Process (by counts) are important. To avoid any confusion, we newly performed gene set enrichment analysis (GSEA) and found the genes responsible for IFN- γ as indicated by "Response To Interferon Gamma", that was significantly induced by DSCR1-knockdown in BV2 cells compared to control. Moreover, these alterations in gene expression were the most significantly enriched biological process in DSCR1-KD BV2 cells by treatment with fA β (**Reviewer Fig. 3**) In both cases, we indeed show that Interferon Gamma signaling play a critical role. For this reason, Interferon Gamma signaling is more likely and we have not mentioned other candidates in this study. **This new data is now added into Revised Supplementary Fig 5. Figure legends and main text are also accordingly modified (Supplementary p.5, line 1-4 and p.11, line 6-12).**

Reviewer Figure 3. Comparison of the significance of the top 10 BPs in control and DSCR1 knockdown BV2 cells with or without fAβ treatment for 24 h (A) The top 10 Gene Ontology (GO) biological processes in DSCR1 shRNA BV2 cells as compared to control shRNA BV2 cells (B) The top 10 Gene Ontology (GO) biological processes in shDSCR1 BV2 cells as compared to shControl BV2 cells with fAβ.

Minor points:

- Characterization of microglia is rather simplistic and incomplete. Besides analyzing Iba1, other DAM microglia markers, such as Trem2 or Clec7a should be examined to substantiate the main conclusion.

Thank you for the helpful comments. Following the reviewer's suggestion, we determined up-regulated Trem2 expression around the plaques in the DSCR1 KO/5xFAD mice compared to 5xFAD control littermates (**Reviewer Fig. 4**). This new data is now added into Revised Fig 3E, F. Figure legends and main text are also accordingly modified (p.32, line 6-15 and p.9, line 1-2).

Reviewer Figure 4. (A) Representative images of Aβ (MOAB2), Trem2 and microglia (Iba1) in 5xFAD mice and DSCR1 KO/5xFAD mice at 5 months. Scale bar = 10 μm. (B) Quantification of microglial Trem2 intensity (5xFAD, n = 499 Iba1-positive (Iba1⁺) cells; DSCR1 KO/5xFAD, n = 399 Iba1⁺ cells). (B) Values are shown as mean ± SEM and tested for statistical significance using a two-tailed unpaired t-test. ****P < 0.0001.

• Despite the claim that DSCR1^{-/-}5xFAD brain contains significantly reduced amyloid plaques (Fig 1), it does not seem that the plaques are smaller (Fig 3a). Is it so? The authors need to quantify and clarify this point

We thank the reviewer to point this out. We have confirmed the reduced size of amyloid plaques in DSCR1 KO/5xFAD mice of the whole brain and cortex but not in the hippocampus (Reviewer Fig. 5). Please check out Revised Fig 1H-J. Figure legends and main text are accordingly modified (p.29, line 3-4 and p.6, line 11-12).

Reviewer Figure 5. (A) Quantification of amyloid plaque size in 5xFAD and DSCR1 KO/5xFAD mice of the whole brain (B), hippocampus (C), and cortex (D) region. (5xFAD; DSCR1 KO/5xFAD mice, n = 4) (A–C) Values are shown as mean ± SEM and tested for statistical significance using a two-tailed unpaired *t*-test. N.S. = not significant, **P* < 0.05, ***P* < 0.01.

Reviewer #2 (Comments to the Authors (Required)):

This report provides data supporting DSCR1 (RCAN1) as a potential therapeutic target in AD. The authors showed that KO of the gene substantially decreased plaque load in the 5XFAD amyloid transgenic. The suggest that his is due to increased uptake and processing by microglia through activation of the endocytic-lysosomal system based on lack of change iAPP mRNA level,, increased Iba1 and LAMP1, CD68 staining and improved lysosomal protein expression in BV2 cells with shRNA knockdown. The authors propose that DSCR1 is a critical mediator of microglial degradation of amyloid plaques and a new potential microglial therapeutic target.

The data on the KO are novel and the effect on plaque load is dramatic and well worth reporting. However, the strength of the evidence attributing this solely to microglial function changes remains too limited to support the specific claims made in the results:

We thank Reviewer 2 for the positive and constructive comments for the improvement of the manuscript

1. While APP mRNA has been assessed, there is not direct data presented on synthetic rates. The author's paper rests on demonstration that breakdown/clearance is independently responsible. I urge direct measures of synthesis, e.g., with 35Smet pulse-chase (e.g., Savage, J Neurosci, 1998).

Reviewer's comments are well acknowledged. In our mouse core facility, we currently have very limited access to use radioactive substances in mice for the *in vivo* labeling experiment. Therefore, we alternatively performed cycloheximide (CHX) chase assay to directly measure synthesis of APP. Following cycloheximide treatment, we examined the half-life of hAPP in the brain of 5XFAD mice and DSCR1 KO/5XFAD mice for different durations. In this new analysis, we showed that the turnover of hAPP was comparable between two groups (5XFAD mice versus DSCR1 KO/5XFAD mice) (Reviewer Fig. 6). This new data is now added into Revised Supplementary Fig 1. Figure legends and main text are also accordingly modified (Supplementary p.1, line 1-5 and p.7, line 16-19).

Reviewer Figure 6. Turnover of hAPP in the hippocampus isolated from 5XFAD and DSCR1 KO/5XFAD mice brains following cycloheximide treatment. (A) Western blot analysis of hAPP after indicated times of protein synthesis inhibition with cycloheximide (CHX). (B) Quantification of APP turnover in the 5XFAD and DSCR1 KO/5XFAD mice. hAPP levels were normalized to Gapdh. (5x FAD; DSCR1 KO/5x FAD mice, n = 3 for each time point)

2. Attribution of all of the relevant effects to microglia is unwarranted. DSCR1 is widely expressed. Other hypotheses are suggested, e.g., could enhanced clearance occur at the BBB with changes in endothelial cell function?

As the reviewer pointed out, DSCR1 might influence several ways of A β clearance in the brain via proteolytic

degradation or the blood–brain barrier (BBB), interstitial fluid (ISF) bulk flow, and cerebrospinal fluid (CSF) egress pathways (13). Based on others and our study, microglia play an important role in the clearance of A β in the brain of 5xFAD mice, and improving microglial phagocytic activity against A β alleviates the amyloid-beta pathology (14). Therefore, we confirmed that DSCR1 ablation showed a limited effect on the BBB permeability by a fluorescence-labeled dextran permeability assay (Reviewer Fig. 7). Thus, it is plausible to suggest that the microglial clearance system plays a pivotal role in our experimental setting. Nevertheless, we would further examine the other possibilities for the clearance of A β in DSCR1 KO mice in a separate study. **This new data is now added into Revised Supplementary Fig 2. Figure legends and main text are also accordingly modified (Supplementary p.2, line 1-5 and p.8, line 7-11).**

13. A systemic view of Alzheimer disease — insights from amyloid- β metabolism beyond the brain. *Nature Reviews Neurology* volume 13, pages612–623
14. TREM2 Lipid Sensing Sustains the Microglial Response in an Alzheimer’s Disease Model. *Cell*, 160, ISSUE 6, P1061-1071,1

Reviewer Figure 7. BBB permeability evaluation in WT and DSCR1 KO mice. (A) Representative images of BBB permeability from the leaking of dextran into the parenchymal space of 3-month-old WT and DSCR1 KO mice. Scale bar = 50 μ m. (B) Relative leakage was quantified by comparing the Dextran-positive fluorescence area. (WT; DSCR1 KO mice, n = 4) (B) Values are shown as mean \pm SEM and tested for statistical significance using a two-tailed unpaired t-test. N.S. = not significant.

3. The evidence for microglial activation for phagocytosis seems incomplete. Markers of activation were limited. Data concerning DAM gene (Keren-Schaul, *Cell*, 2017) expression in the microglia would add meaningfully to the case.

We thank to the reviewer for this helpful suggestion. We added the new data of immunohistochemistry for Trem2. Please check out the Reviewer’s figure 4.

Reviewer Figure 4. (A) Representative images of A β (MOAB2), Trem2 and microglia (Iba1) in 5xFAD mice and DSCR1 KO/5xFAD mice at 5 months. Scale bar = 10 μ m. (B) Quantification of microglial Trem2 intensity (5xFAD, n = 499 Iba1-positive (Iba1⁺) cells; DSCR1 KO/5xFAD, n = 399 Iba1⁺ cells). (B) Values are shown as mean \pm SEM and tested for statistical significance using a two-tailed unpaired t-test. ****P < 0.0001.

4. Finally, as minor details of interest, can the authors augment their descriptions of plaque morphology? E.g., are dense core plaques increased?

We thank to the reviewer for this helpful suggestion. Following the reviewer's suggestion, we performed a new experiment showing significantly fewer dense core plaques in the DSCR1 KO/5xFAD mice by staining with Thioflavin-S (Reviewer Fig. 8). Please check out Revised Fig 1K-N. Figure legends and main text are accordingly modified (p.29, line 6-15 and p.6, line 12-14).

Reviewer Figure 8. (A) Representative images of Thioflavin-S positive A β plaques in 5xFAD and DSCR1 KO/5xFAD mice at 5 months. Scale bar = 500 μ m (B) Quantitative analysis of Thio-S positive amyloid plaques area in 5xFAD and DSCR1 KO/5xFAD mice of the whole brain (B), hippocampus (C), and cortex (D) region. (5xFAD; DSCR1 KO/5xFAD mice, n = 4) (B–D) Values are shown as mean \pm SEM and tested for statistical significance using a two-tailed unpaired t-test. *P < 0.05, **P < 0.01, ***P < 0.001

November 16, 2022

RE: Life Science Alliance Manuscript #LSA-2022-01556-TR

Prof. Jiyoung Park
Ulsan National Institute of Science and Technology
Biological Sciences
50 UNIST-gil
Ulsan, NA 44919
Korea, Republic of (South Korea)

Dear Dr. Park,

Thank you for submitting your revised manuscript entitled "DSCR1 knockout ameliorates the A β pathology of Alzheimer's disease by enhancing microglial activity". We would be happy to publish your paper in Life Science Alliance pending final revisions necessary to meet our formatting guidelines.

- please upload both your main and supplementary figures as single files
- please use the [10 author names, et al.] format in your references (i.e. limit the author names to the first 10)
- please add your supplementary figure legends and your table legends to the main manuscript text
- please upload your table files as editable doc or excel files or make sure that the tables are part of the doc file of your main manuscript

Figure Check:

- Figure 4A: there are splices in the middle of the figure that need to be noted by a vertical solid line

A. FINAL FILES:

B. MANUSCRIPT ORGANIZATION AND FORMATTING:

Sincerely,

November 21, 2022

RE: Life Science Alliance Manuscript #LSA-2022-01556-TRR

Prof. Jiyoung Park
Ulsan National Institute of Science and Technology
Biological Sciences
50 UNIST-gil
Ulsan, NA 44919
Korea, Republic of (South Korea)

Dear Dr. Park,

Thank you for submitting your Research Article entitled "DSCR1 deficiency ameliorates the A β pathology of Alzheimer's disease by enhancing microglial activity". It is a pleasure to let you know that your manuscript is now accepted for publication in Life Science Alliance. Congratulations on this interesting work.

DISTRIBUTION OF MATERIALS:

Again, congratulations on a very nice paper. I hope you found the review process to be constructive and are pleased with how the manuscript was handled editorially. We look forward to future exciting submissions from your lab.

Sincerely,
